# Requirements for Drone Operations to Minimise Community Noise Impact

**DOI:** 10.3390/ijerph19159299

**Published:** 2022-07-29

**Authors:** Carlos Ramos-Romero, Nathan Green, Seth Roberts, Charlotte Clark, Antonio J. Torija

**Affiliations:** 1Acoustics Research Centre, University of Salford, Manchester M5 4WT, UK; c.a.ramosromero@salford.ac.uk (C.R.-R.); n.green7@edu.salford.ac.uk (N.G.); 2Hayes McKenzie Partnership Ltd., Unit 3 Oakridge Office Park, Salisbury SP5 3HT, UK; seth@hayesmckenzie.co.uk; 3Population Health Research Institute, St George’s University of London, London SW17 0RE, UK; chclark@sgul.ac.uk

**Keywords:** drone noise, community noise impact, noise annoyance, sleep disturbance, noise regulation, noise metrics

## Abstract

The number of applications for drones under R&D have growth significantly during the last few years; however, the wider adoption of these technologies requires ensuring public trust and acceptance. Noise has been identified as one of the key concerns for public acceptance. Although substantial research has been carried out to better understand the sound source generation mechanisms in drones, important questions remain about the requirements for operational procedures and regulatory frameworks. An important issue is that drones operate within different airspace, closer to communities than conventional aircraft, and that the noise produced is highly tonal and contains a greater proportion of high-frequency broadband noise compared with typical aircraft noise. This is likely to cause concern for exposed communities due to impacts on public health and well-being. This paper presents a modelling framework for setting recommendations for drone operations to minimise community noise impact. The modelling framework is based on specific noise targets, e.g., the guidelines at a receiver position defined by WHO for sleep quality inside a residential property. The main assumption is that the estimation of drone noise exposure indoors is highly relevant for informing operational constraints to minimise noise annoyance and sleep disturbance. This paper illustrates the applicability of the modelling framework with a case study, where maximum A-weighted sound pressure levels $LA_{max}$ and sound exposure levels SEL as received in typical indoor environments are used to define drone-façade minimum distance to meet WHO recommendations. The practical and scalable capabilities of this modelling framework make it a useful tool for inferring and assessing the impact of drone noise through compliance with appropriate guideline noise criteria. It is considered that with further refinement, this modelling framework could prove to be a significant tool in assisting with the development of noise metrics, regulations specific to drone operations and the assessment of future drone operations and associated noise.

## 1. Introduction

One of the most important recent changes in the civil aviation industry is the imminent incorporation of unmanned aerial vehicles (UAVs) into operations, especially for transportation and logistics. The growing interest in UAVs (commonly referred to as drones) is due to their technical, operational and economic benefits such as last-mile transportation, quick medical deliveries to both urban and remote areas, and the important reduction in overall CO_2_ footprint and local air quality emissions [1,2,3].

However, whilst the broad range of potential applications of drone technologies could bring substantial benefits, they can also produce new and unconventional sources of environmental noise, which are likely to lead to significant challenges for the health, quality of life, and well-being of exposed communities. In addition, UAVs are expected to operate differently than the standard aircraft. UAVs will operate closer to ground level and in proximity to dwellings, especially during the last-mile manoeuvres such as take-off, landing and hovering. It is important to take care not only of the outdoors environmental drone noise distribution near the flight path but the noise levels on the façade and the noise levels transmitted inside the buildings [4]. The literature on drone noise emission has reported that drone noise signature is highly influenced by the type of drone configuration, e.g., number of rotors in multi-copters, size and weight, flight manoeuvres [5,6,7,8], and also significantly influenced by the ambient weather conditions, most notably the wind [9].

The main aim of this paper is to present a modelling framework for setting up best operational practices for drone manoeuvres to minimise the potential adverse impacts on receivers inside buildings. The importance of this framework is that the drone stakeholders can be informed of the specific requirements for distances from residential properties on the basis of noise metrics specified as guidelines for acoustic targets in the receiving environment, as depicted in Figure 1. The design of the framework is presented in detail and illustrated with a case study which uses data from actual outdoor measurements and models the outdoor sound propagation and sound transmission façade features to predict noise exposure inside a sensitive room that can be compared against relevant noise criteria.

The noise emission of conventional aircraft noise sources is usually reported by maximum A-weighted sound pressure levels, $LA_{max}$; sound exposure level, SEL; sound levels integrated over a day/period time such as $LA_{eq,T}$; and spectral analysis by 1/*n* octave bands. However, these noise metrics have been found unable to account for the acoustic characteristics of drone noise due to their substantial content in complex tones and high-frequency broadband noise [10,11]. For instance, as described by Torija et. al. [12], the tonal correction factor in the effective perceived noise level (EPNL) metric, used for aircraft noise assessment, might not be suitable for capturing the perceptual effect of complex tonality. To date, enough evidence has been found demonstrating that the noise produced by drones does not qualitatively or quantitatively resemble the noise produced by conventional aircraft [12,13,14].

The inability of existing aircraft noise metrics for capturing drone noise perception might suggest that the current evidence for human responses to aircraft noise is not applicable for drone noise exposure. Sound quality metrics (SQM) could outperform traditional noise metrics in capturing the key psychoacoustic factors influencing perception of drone noise. This is discussed by Torija and Clark [15], where the authors suggest that further research is needed to define metrics optimised for drone noise and also to define acceptable levels for drone noise emission. However, until enough robust evidence on the human response to drone noise exposure is gathered, existing noise metrics and recommended target could be used to inform regulation of operational procedures. This assumption is also supported by some recent research suggesting loudness-related metrics as the main drivers of noise annoyance for drone operations [7,16]. 

The modelling framework presented in this paper therefore relies on existing evidence for aircraft noise exposure (e.g., WHO guidelines for sleep disturbance and awakenings [17]) and current aircraft noise metrics. However, other acoustics metrics have been developed, for example based on SQM, that have proven to be useful in the analysis of perceptions of drone noise and other conventional urban noise sources [14]. 

The paper is organized as follows: Section 2 describes the drone noise database used for the case studies; Section 3 describes the modelling framework to set the distance requirements for drone operations; Section 4 presents the results of the framework applied to three case studies; Section 5 discusses the framework application and future work; and the main conclusions of this work follow in Section 6.

## 2. Drone Sound Signals Database

The research presented in this paper relies on the analysis of an acoustics database, reported by Volpe [18], that includes three types of multirotor and fixed-wing uncrewed aircraft on several flying operations (hovering, flyover, take-off, landing, and facilities inspection). Table 1 presents the main design specifications of the tested multirotor aircraft. The list includes drones of different weights and dimensions and the aircraft ground speed of the flyover tests for each drone.

Figure 2 illustrates the setup used for measurements that have been analysed within the Volpe database with an inverted centreline ground (CLG) microphone below the drone flight path 150 feet above the ground (~47.5 m). The drone noise database provides the sound pressure levels only at distances r, equal to or longer than the slant distance $r_{sd}$ (shortest straight line between the microphone and the sound source equal to the height above ground).

From the acoustic data and audio files provided within the acoustic measurement database for flyover operations at slow and fast speeds, it is possible to visualize the spectral content of the tested drones. The typical acoustic footprint of this type of vehicle is produced by the rotor and propellers [19], and it is characterised by the tonal components, which include the effects of the rotors’ fundamental (low frequencies), the harmonics of the rotors’ fundamental frequencies (mid), and the electric motor noise component (high frequencies) [20]; see Figure 3. One interesting observation about the measured data is that the tonal components appear to be more significant in terms of amplitude when the drone is performing a fast flyover.

## 3. Framework for the Drone Operations Requirements

The gradual incorporation of drones into soundscapes near communities highlights the need to develop new policies and engineering tools to deal with the potential noise impacts of these new sound sources on exposed populations. In this regard, the drones’ sound emissions need to comply with environmental recommendations to minimise the potential health effects of noise. Several constraints for drone operations could be considered, i.e., distance from residential properties during drone flyover, along with speed and/or altitude.

Using the core process outlined in the flow chart in Figure 4, indoor sound levels can be estimated from drone noise (generated outside during the flying operation) by simulating the typical transmission loss during the propagation from the drone to immediately outside the building façade and then the transmission through the façade into the receiver room. Once the sound levels indoors are estimated, the drones’ operational constraints can be set to comply with the guidelines of the acoustic objective on the receiver side.

In particular, this research paper provides the preliminary results of the application of the proposed framework to establish the minimum drone/façade distance DFd. This can be studied as an optimisation parameter on drone infrastructure path planning [21]. If a minimum DFd parameter can be established, it can be controlled by the drone operator or pilot to minimize the noise impact inside the receiver room accordingly. DFd  was calculated using current WHO night noise guidelines [17]. The more recent WHO Environmental Noise Guidelines for the European Region are complimentary to the earlier WHO Noise Guidelines and do not include any updated recommendations for single noise events. 

### 3.1. Outdoor Sound

The sound pressure level of an operative drone flying outdoors is a function of the environmental conditions and the flight manoeuvre variables. Therefore, the produced noise amplitude for specific drone models can be obtained by actual measurement campaigns either on field or in laboratory conditions. Acoustic propagation models can also be included in the analysis to report the likely effects of changes in environmental and operational conditions, such as the temperature, humidity, and source/target distance.

From the extensive field-based drone measurement campaign reported by [18], it is possible to obtain the actual sound levels during the drone flyover at microphone distance r. The drones’ operations measured and presented within this campaign were flyover (both fast and slow), take-off, landing, hovering, and infrastructure inspections.

To obtain the sound pressure level at distances other than the measurement drone-receiver slant distance (*r*), state-of-the-art sound propagation models can be applied to estimate the sound level at other distances under varying environmental conditions needed in the analysis. Previous research, identified during the literature review, recommended some approaches to outdoor drone noise propagation. The impact of drone noise during hovering operations in large outdoor urban environments has been explored using acoustic ray tracing. The ground reflection and acoustic refraction by an inhomogeneous atmosphere were included in the sound propagation model [22]. 

Moreover, several parameters conform to a comprehensive model of sound propagation, included as modifying factors $A_{i}$ in the generic expression of sound pressure Level, $L_{p}$, in terms of the sound power level, $L_{w}$ (Equation (1)) [23]. This approach has been implemented in commercial software for sound propagation according to methodology within the ISO 9613-2 standard [24]:(1)$$L_{p}=L_{w}+\Sigma_{i}A_{i}$$

Two main assumptions have been considered in this paper for the selection of the propagation modifying factors. Firstly, the drone is a single-point sound source propagating with spherical spreading characteristics [19,22], and the operation of the drone’s electric motors generates a significant noise with high-frequency components which, unlike the acoustic signature of typical aircraft noise, reach the receiver without significant attenuation due to drones operating at shorter source/receiver distances [20]. The spherical sound field can be calculated from the sound power level of the source, considering the geometrical attenuation due to spherical spreading and the attenuation associated with atmospheric absorption [25]. These two parameters are presented in Equation (2), with distance r and the atmospheric sound absorption coefficient α as the main modifying factors during the drone sound propagation effects:(2)$$L_{p}\left(r,\alpha \right)=L_{w}+A_{r}+A_{\alpha ,r}$$

The effect of the mentioned contributors at a reference distance $r_{m}=r_{sd}$ from the drone are included in the measurements of $L_{m}$, and the sound power $L_{w}$ can be derived through the calculation of the contributors as presented in Equation (3):(3)$$L_{w}=L_{m}-A_{r_{m}}-A_{\alpha ,r_{m}}$$

Therefore, the sound pressure level at any distance r, which is obtained by Equation (4), is usually r>1 m to avoid the effects of the near field [26]. The effects on the atmospheric modifying factor on sound level attenuation due to the atmospheric sound absorption are included in the term $A_{\alpha ,r}=L_{atm}$, where α is the attenuation coefficient for air absorption in Equation (5) [27,28]:(4)$$L_{p}\left(r\right)=L_{m}-20\log_{10}\left(\frac{r}{r_{m}}\right)-L_{atm}$$
(5)$$L_{atm}=10\log_{10}e^{2\alpha r}$$

Furthermore, if the estimation of the $L_{p}$ is based on the maximum sound pressure level, i.e., $L_{m}=LA_{max}$ the sound exposure level SEL can be obtained with an effective time $t_{e}$ by Equation (6) [27,29]:(6)$$SEL=LA_{max}+10\log_{10}\left(\frac{t_{e}}{t_{0}}\right);t_{0}=1\;s.$$

The sound signal on the time domain presented in Figure 5 shows the amplitudes of $LA_{max}$ and $LA_{eq}$ registered every 0.5 s [18]. It is noted that the two values are almost equal during a fast flyover operation for each of the three drones that were measured. However, this condition may not be present for other drone manoeuvres. The accelerating and deaccelerating drones’ mechanisms are mainly related to the rotational speed and tilting rotor position but produce changes in the pitch of the emitted sound; therefore, the drone operations can cause both spectral content and time variations of sound with consequences in the drone acoustics noise impact [30] and annoyance [16]. 

### 3.2. Sound Transmission

The next step in the sound path from outdoors to the receiver environment is the effects of acoustic attenuation due to a building façade. The sound reduction properties of the façade can be obtained either through experimental measurement or modelling, although the sound reduction properties resulting from these two methods, even for the same building element, may differ significantly. The sound reduction index R is derived by measuring the difference between the sound levels at the source ($L_{S}$) and the receiver ($L_{R}$), which are separated by a partition element with surface area S. The equivalent sound absorption area provided by the receiving room $A_{R}$ is considered in the calculation of R, as is presented in Equation (7) [31]:(7)$$R=L_{S}-L_{R}+10\log_{10}\left(\frac{S}{A_{R}}\right)$$

The experimental results of the measured sound reduction indexes depend strongly on the spectral content of the sound within the source room, building materials, glazing area, open-window condition, and receiver room measurement positions. The contribution of different sound wave paths (such as flanking transmission) can be measured during experimental façade testing. The cumulative sound reduction through all transmission paths can be expressed using the apparent sound reduction index $R^{\prime }$ [31,32,33,34].

Based on the preliminary analysis of the drone sound database, this paper will focus on the outdoor flyover procedures at fast and slow speeds. From a noise perception point of view, the acoustic signatures of the recorded drones can be particularly interesting because the frequency bands with the highest amplitudes could potentially overlap with frequencies of reduced sound reduction performance as a result of the resonance or coincidence frequencies of the partition materials. The sound emission of drone flyover and both the modelled and measured sound reduction indices for a typical closed double-glazed residential window are presented in Figure 6.

Bot, the experimentally reported apparent sound reduction index (R′) and commercial software prediction (sound reduction index R) for a typical double-glazing façade-window configuration [32] decreases or dips in the sound reduction index at low-frequency (~250 Hz) and high-frequency bands (~3150 HZ) where the coincidence dip would be expected, whereas the $LA_{max}$ values of a flyover operation (Yuneec Typhoon) shows the highest amplitudes in these same frequency bands. 

This overlap between the reduced noise attenuation provided by the window and the higher third-octave band amplitudes of the drone-source acoustic signature presents an interesting and potentially problematic unexplored area with relevant applications for the regulations and requirements regarding drones operating near to communities.

The expression in Equation (8) lets us estimate the sound pressure level in the receiver environment with volume V partition surface area S  and reverberation time T, as described by Equation (4) of the standard BS EN ISO 12354-3:2017 [31].
(8)$$L_{indoors}=L_{outdoors,2m}-R^{\prime }+10\log_{10}\left(\frac{TS}{0.16V}\right)$$

### 3.3. Indoor Sound

The impact of drone noise on indoor receivers is a current gap in the research because the techniques for noise evaluation applied for other noise sources (i.e., traffic noise and aircraft noise) are based on the overall emission levels, and the main amounts of energy in their sound signal are located in the low and middle broadband ranges. The special acoustics characteristics of drone noise signature; in particular, the pure tones and high-frequency broadband noise means that traditional sound level metrics may not sufficiently represent the acoustic signature of drones or effectively convey their noise impact.

However, acoustic screening based on the overall sound levels considered in the noise assessment guidelines is an important starting point to assess the possible negative effects that the drone noise source may have on an exposed community [35]. For instance, the WHO guidelines for sleep quality effects can be used to set these acoustic requirements [17]. These WHO guidelines set the threshold of the $LA_{max,inside}\;$ at 42 dB for “Waking up at night and/or too early in the morning”. In addition, the number of drones in operation could be considered a variable to adjust to model recurrent flight events and predict the exposure to drone noise over an appropriate time period, e.g., 8 h at night.

This example maximum noise level criterion is helpful in determining a suitable minimum DFd  during the night hours when sleep disturbance is an important factor for determining impact. However, this criterion alone does not provide any insight into the number of drone passbys that might be acceptable or provide any idea of acceptability during daytime hours. Currently in the UK, commercial aircraft noise is assessed based on government guidance about the lowest observable adverse effect level (LOAEL), quantified in terms of the $LA_{eq,16\mathrm{hour}}$ and $LA_{eq,8\mathrm{hour}}$ for day and night respectively, as calculated for a given airport over an “average summer day” (defined over a standardised 92-day period between 16th of June and 15th of September). The government guidance on the LOAEL values for day and night is informed by the 2014 Survey of Noise Attitudes (SONA), the results of which are published in CAP1506 [36].

As discussed by Torija and Clark [15], it is not clear whether or not current metrics or indeed criteria such as the LOAEL for commercial aircraft noise are suitable for assessing drone noise at residential receiver locations. However, the latest best practice for assessing commercial aircraft noise acknowledges that average noise metrics such as the $LA_{eq}\;$ may not adequately represent what is actually perceived at receiver locations, particularly those close to an airport where the noise environment is punctuated by very loud individual flyover events rather than a continuous drone of distant aircraft noise. 

In order to account for this, number above metrics such as the NA70 are often used as secondary metrics to help describe the number of times $LA_{max}$ exceeds a given level; in the case of NA70, this would be the number of events above 70 dB $LA_{max}$ at a given receiver location. It is noteworthy that NA70 has been used as a national indicator in Australia for potential speech intelligibility issues with respect to commercial aircraft noise on the basis of an assumed 70 dB $LA_{max}$ outside a residential location (as discussed in Appendix B of CAP1506 [36]). In the absence of anything specifically derived for drone noise, a daytime maximum noise criterion of 70 dB $LA_{max}$ would be an appropriate starting point for deriving a minimum daytime DFd  that might reasonably be expected to avoid speech intelligibility issues.

Further research is needed to be able to derive suitable limits for either the number of drone flyover events above a certain maximum level, or an average noise metric that can be shown to correlate with mean annoyance (as is the case for $LA_{\mathrm{eq},16\mathrm{hour}}$ as discussed in CAP1506).

## 4. Results—Case Studies

In this paper, we report on the recommendations for drone/façade distance DFd during flying operations based upon actual measurements of drones operating in an outdoor environment. In this case, the proposed framework establishes the receiver noise level criterion to comply with the WHO guidelines for sleep quality based upon $LA_{max,\;indoors}$.

The flying conditions remained stable during the whole exercise, i.e., flight path and speed of the drone. Firstly, the drone is considered a noise source operating outdoors. Then, the amplitude of the indoors sound was estimated from a façade configuration located at the source-receiving interface. Finally, the recommendations for DFd are based on the $LA_{max,indoors}$. The developed modelling framework is illustrated considering the specific acoustic and operating conditions of three different drones listed in Table 1 during fast and slow speed flyovers.

Figure 7 shows the amplitude of $LA_{max}\;$ as a function of the DFd in the receiver environment without (left) and with (right) the hypothetical installation of the partition with the window considering an open area, in this case, $0.05\;m^{2}$.

Data for the assumed façade installation have been presented in third-octave band R′, which is representative of a standard glazing element with a partially open window. The partition’s sound reduction performance (inward lateral rotation window with standard glazing type 4-16-4 mm) was measured under controlled conditions and reported by Waters-Fuller and Lurcock [32]. The predicted sound signal at the receiver shows that the attenuation at high frequencies is not great considering the significant emission of the drone in this range of frequencies.

The waterfall diagrams in Figure 7 present the sound amplitudes ($LA_{max}\;)$ obtained at a distance r from the source, both by actual measurements ($r\geq r_{sd}$) and extrapolated through predictions from modelled propagation ($1\;m<r<r_{sd}$). The shortest drone-sensor distance $r_{sd}$ is highlighted. Then, it is possible to obtain the broadband $LA_{max}\;$ through the signal amplitude on each frequency band.

Finally, the recommendations for the drone operations variables can be compared with the acoustic target, and it is possible to estimate the compliance conditions in the receiver environment.

The recommended DFd was obtained from fitting curve modelling based on $LA_{max,indoors}$ associated with different flyover operations, drone types, and closed-window configurations. 

A fitting curve model (Figure 8) was estimated to establish a recommended drone operation parameter. Therefore, it is possible to obtain a fitting curve specific to each combination of drone flyover operation and building façade configuration to achieve the indoor noise criterion/acoustic target.

Consequently, it is possible to find the recommended drone operational parameter to comply, for instance, with the sleep quality guidelines (42 dB $LA_{max,indoors}$) or biological effects on EEG awakening (35 dB $LA_{max,indoors}$) as reported by WHO [17]. 

Next, some recommended DFd values were obtained from the proposed modelling framework. The results let us compare the recommended DFd by considering the type of drone, the speed of the flying operation, and the open/closed window configuration. 

### 4.1. One Drone—Open Window Conditions–Fast/Slow Speed Flyover

The broadband sound emission of the contra-rotating octocopter GD28X was analysed as a function of the flyover speed. The recommended Drone/façade distance for both fast and slow flyovers is presented in Table 2**,** where a partially open window ($0.05\;m^{2}$) was considered for the glazing configuration. To comply with the acoustic guideline, the approximate increase of 5 m/s in the speed during flying over operations needs to consider an increment of DFd on 68.5 m if the receiver window is partially opened, as is depicted in Figure 9. For this type of aircraft, the DFd greater than 110 m was recommended to ensure the acoustic objective of 42 dBA indoors during the studied overflights.

### 4.2. Three Drones—One Window Condition

Table 3 presents the drone/façade distance DFd recommendations for the three tested drones at their own fastest speed during flyover when the receiver window is closed. The indoor acoustic target would be achieved with distances less than 25 m for the aircrafts M200 and Typhoon, while distances higher than 60 m would need to be defined for the drone GD28X.

Although the available flyover speed data for each drone do not let us compare the operations at the same speed, the values of $LA_{max,indoors}$ from each drone at fast flyover speed would be a consequence of the construction characteristics of the drones, i.e., weight, size and number/type of rotors (see Table 1).

From this preliminary observation, drones with smaller dimensions, lighter total take-off weight, and fewer rotors could operate closer to the community than drones with larger proportions and more motors to comply with the acoustic target indoors, as is depicted in Figure 10.

### 4.3. Drone Fast Flyover Operation and Open Window Condition

When comparing the different open-window configurations during one drone overflight, the acoustic target indoors can be obtained by increasing the drone/façade distance DFd if the open window area is increased, as presented in Table 4, where the results are based on the drone Typhoon operations. Furthermore, doubling the open area, the drone path should move away from the facade between 10 and 13 m to ensure the acoustic metric target at the receiver.

For this specific glazing configuration, the closed window can reduce the sound by 30 dB compared with the fully open window. Therefore, the distance at which the drone could operate to maintain the acoustic target (42 dBA) between the fully open and closed window conditions shows a difference of 115 m. See Figure 11 (left).

The analysis with the same previous conditions of both flyover drone and window configurations can be depicted to compare the acoustic target based on $SEL_{inside}$ as is presented in Figure 11 (right).

## 5. Discussion

The modelling framework presented in this paper can contribute to setting operational requirements for drone operations to protect communities exposed to their noise. The outcomes of this modelling framework could be used in multi-criteria decision-making for trajectory optimization [37]. This allows components from the source to the propagation path for indoor sound transmission to be updated independently from one another. For instance, in the case studies presented in Section 4, the source data are based on drone measurements outdoors. Instead, the source emission could be based on acoustic predictions or measurements carried out in anechoic chambers or wind tunnels. The sound propagation from source to building façade only accounts for spherical divergence and atmospheric absorption. However, the propagation model can be expanded to account for other propagation factors such as ground effects, reflection from surfaces, and screening by obstacles according to ISO 9613-2. The sound transmission through building partitions is based on data gathered in a comprehensive measurement campaign with actual glazing installations [32]. However, input data from modelling software (e.g., INSUL) for sound transmission can be used to account for other building outdoor/indoor interfaces and window/façade configurations.

For a given drone model, the minimum drone–façade distance to meet the WHO recommendations can vary significantly depending on the glazing condition. This implies the need to accurately define the sound transmission indices for the most relevant glazing system including the window opening behaviour, which varies across individuals and also seasonally. 

Neutral atmospheric conditions are considered in this paper; however, wind speed and direction can significantly influence both the sound emission [27] and propagation to the receiver. Appropriately accounting for these factors is crucial for setting operational requirements representative for the area under study. Finally, the drone is assumed to be an omnidirectional source. This is unlikely, and therefore, the directionality of the drone sound emission should be considered.

The acoustic objectives used in this paper to set operational requirements are based on conventional noise metrics and existing evidence on human responses to aircraft noise. As discussed by Torija and Clark [15], current noise metrics seem to be unable to account for the acoustic characteristics of drones (i.e., complex tonal content, substantial noise emission above 4 kHz), and it is not certain that existing evidence for aircraft noise and health effects can be directly applicable to drone noise. As evidence on human response to drone noise expands, new acoustic objectives can be used in the modelling framework to set requirements for drone operations.

The analysis in this paper is restricted to 1/3 octave banding. This limits the investigation of how the coincidence dips in glazing systems might influence the indoor transmission of the substantial high-frequency noise of drones. This issue can be addressed with more refined frequency analysis, but the lack of narrowband data for sound transmission coefficients is a limiting factor. Further work will be done to account for this limitation.

Due to the frequency content of drones, vibration-induced noise generating at a glazing system is not considered in this paper. However, further extension of the modelling framework for larger vehicles (e.g., urban air mobility vehicles) will consider this sound generation mechanism.

## 6. Conclusions

This paper presents a modelling framework for the estimation of indoor noise exposure due to drone operations. This modelling framework can be used to define operational restrictions (e.g., in the form of drone-façade distance) to meet recommended noise targets and avoid significant noise impacts on communities inside dwellings. The current version of the modelling framework is based on the measured drone sound signature and the sound propagation outdoors. The method also includes the effects of the sound attenuation provided by masonry and glazing elements during the sound transmission into the receiver room.

The application of this modelling framework is illustrated with case studies, where the minimum distance from a given drone to a typical residential building is defined to comply with the noise requirements to avoid sleep disturbance.

The results of the estimation of maximum A-weighted sound pressure levels $L_{Amax}$ and sound exposure levels SEL as received in typical indoor environments are presented as case studies of drone/façade distance recommendations. To do this, a series of drone sounds recorded during outdoor flyover operations in free field and predicted amplitudes from sound propagation models were filtered to simulate the transmission loss through a standard façade configuration.

For a given drone (GD28X) operating near a window with an open area of 0.05 m^2^, the drone-façade distance to meet WHO recommendations for sleep quality: waking up in the night and/or too early in the morning ranges between 110.6 m (slow flyover) and 179.1 m (fast flyover). If one closed-window glazing configuration is considered, the minimum drone–façade distance ranges between 15.8 m (Typhoon) and 62.7 m (GD28X ) to meet the same acoustic target during fast flyovers. Finally, for the Typhoon drone operating at a fast flyover speed, the drone–façade distance ranges between 15.8 m (closed-window) and 131.6 m (fully open/window) to comply with the sleep quality WHO guidelines. 

The practical and scalable capabilities of the modelling framework presented in this paper are part of a strategy to develop a set of tools for inferring and assessing the impact of drone noise through compliance with different regulations and guidelines. This methodological framework can be leveraged by stakeholders in the drone sector for trajectory optimisation for drone vehicles for applications such as parcel delivery. With further refinement, this type of modelling framework has the potential to significantly assist in developing noise metrics, regulations, and guidance specific to drone noise as well as in assessing the noise from future drone operations. In this paper, the objective of the modelling framework is to inform the distance requirements to operate drones avoiding issues due to community noise impact. Further work will expand the capabilities of this modelling framework to improve accuracy and so the framework outputs can be used for auralisation and psychoacoustic analysis.

## Figures and Tables

**Figure 1 ijerph-19-09299-f001:**
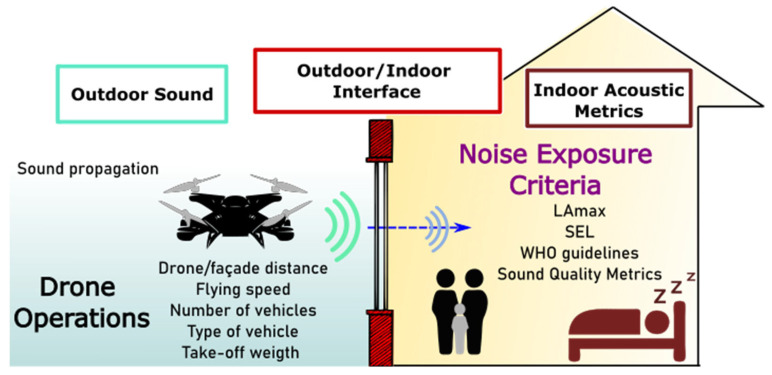
Framework for drone operating recommendations based on acoustic metrics analysis.

**Figure 2 ijerph-19-09299-f002:**
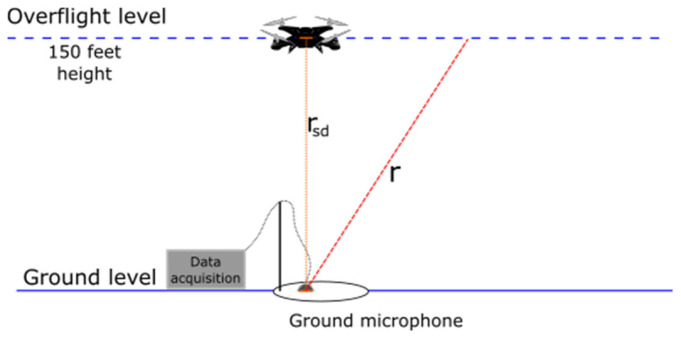
The measurement setup. Adapted with permission from Ref. [18].

**Figure 3 ijerph-19-09299-f003:**
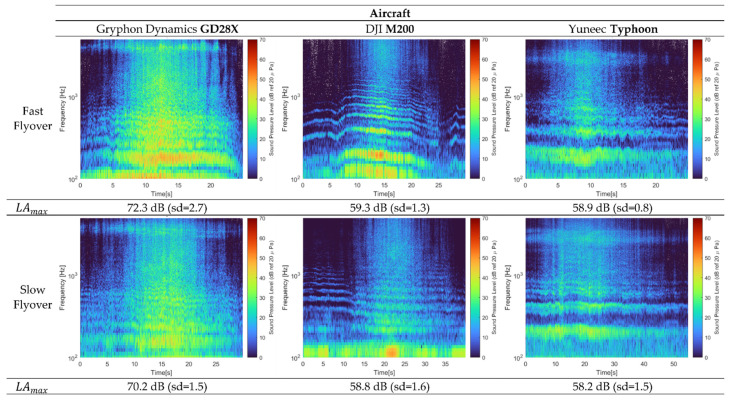
Spectrograms and $LA_{max}$ values for three drone models: GD28X (**left**), M200 (**middle**), and Typhoon (**right**) during fast (**top**) and slow (**bottom**) flyover operations at ~47.5 m altitude above the CLG. Data adapted with permission from Ref. [18].

**Figure 4 ijerph-19-09299-f004:**
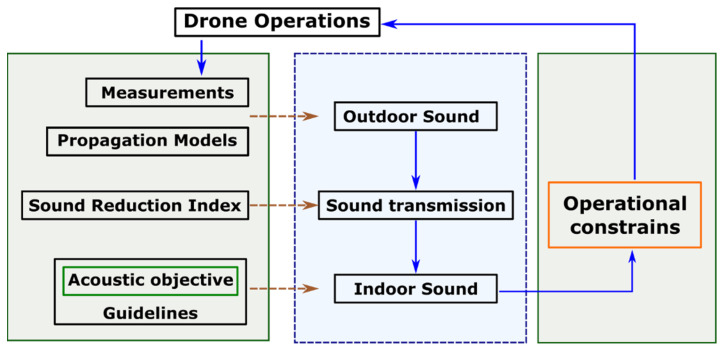
The modelling framework for the requirements of drone operations based on the guidelines for indoor acoustic objectives.

**Figure 5 ijerph-19-09299-f005:**
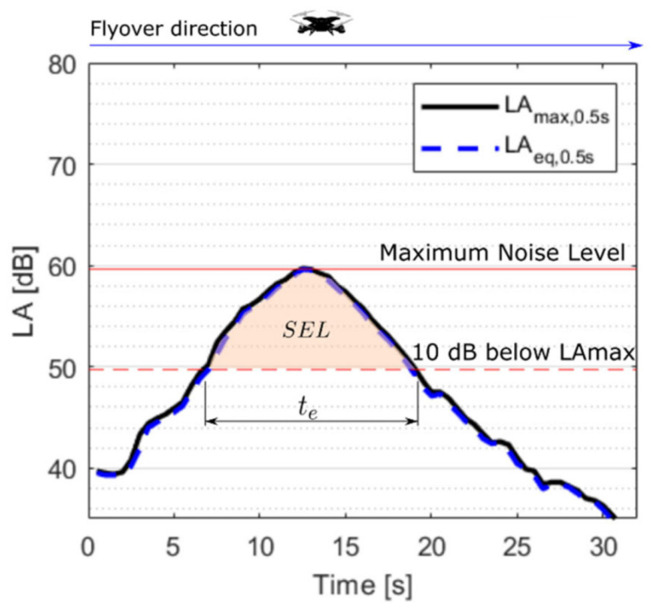
Sound exposure level SEL, maximum noise level $LA_{max}$, and effective time $t_{e}$. The presented data corresponds to a recording during a fast flyover operation of the drone Typhoon.

**Figure 6 ijerph-19-09299-f006:**
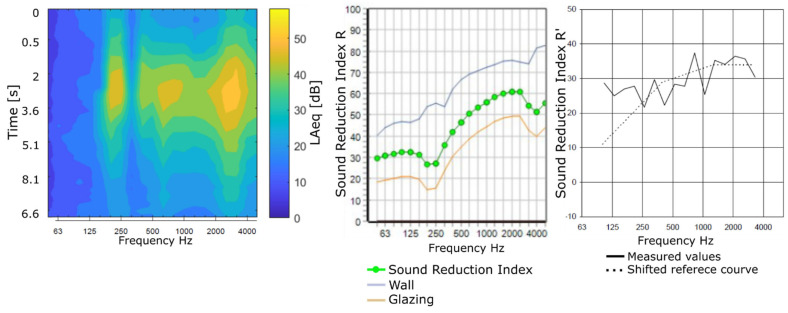
$LA_{max}$ for the drone Typhoon during fast flyover operations (**left**); adapted with permission from Ref. [18]. Sound reduction index of a 70 mm PVC-U bottom hung inward tilt window 4-16-4 mm (0.94 m^2^; 26.00 kg/m^2^). Predicted R by INSUL software (**middle**) and measured $R^{\prime }$ (**right**); reprinted with permission from Ref. [32]. Note that “shifted reference curve” is the specific term used in [31] for obtaining a single value from frequency-dependent sound reduction index.

**Figure 7 ijerph-19-09299-f007:**
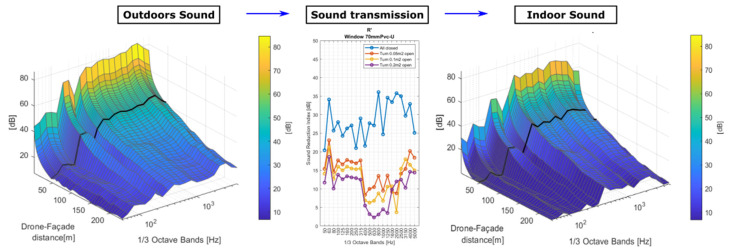
$LA_{max}\;$ by frequency as a function of the drone/façade distance r, outdoors (**left**) and indoors (**right**) through the assumed partition ($0.05\;m^{2}\;$ window open area). The measured apparent sound reduction index $R^{\prime }$ of the partition due the glazing open area is included (**middle**); adapted with permission from Ref. [32].

**Figure 8 ijerph-19-09299-f008:**
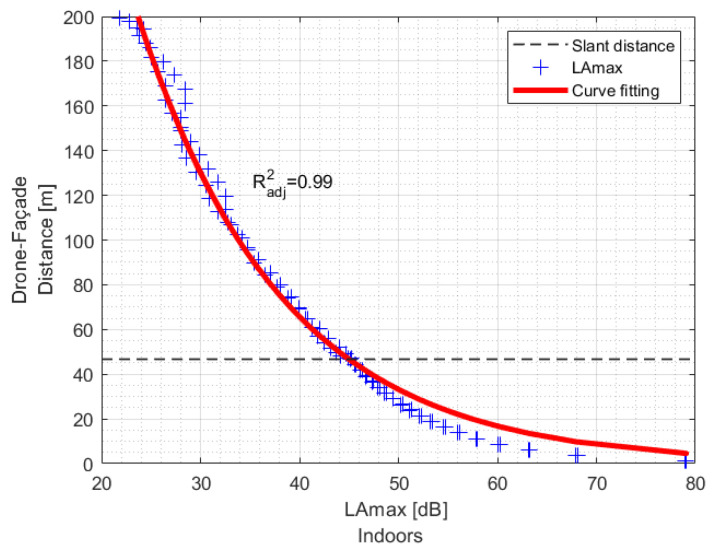
Drone/façade distance DFd as a parameter to estimate fitting curve model as a function of $LA_{max,indoors}$.

**Figure 9 ijerph-19-09299-f009:**
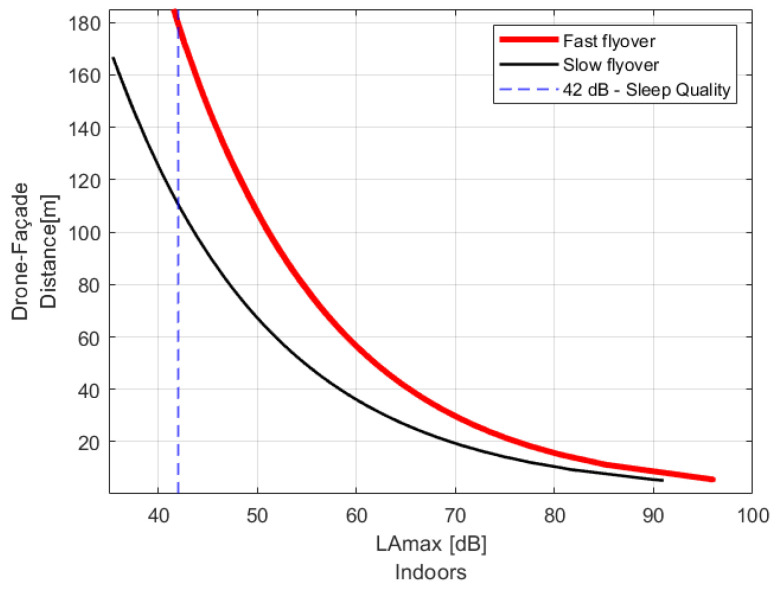
Recommended drone/façade distance based on the $LA_{max,indoors}$ for drone GD28X at fast and slow flyover operation near a façade with a conventional window–open area 0.05 m^2^.

**Figure 10 ijerph-19-09299-f010:**
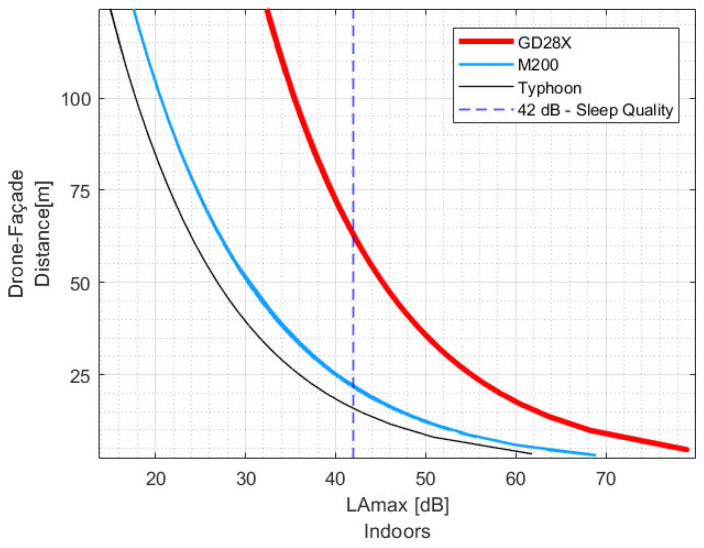
Recommended drone/façade distance based on the $LA_{max,indoors}$ for the tested drones at fast flyover operation near a façade with a conventional window–closed.

**Figure 11 ijerph-19-09299-f011:**
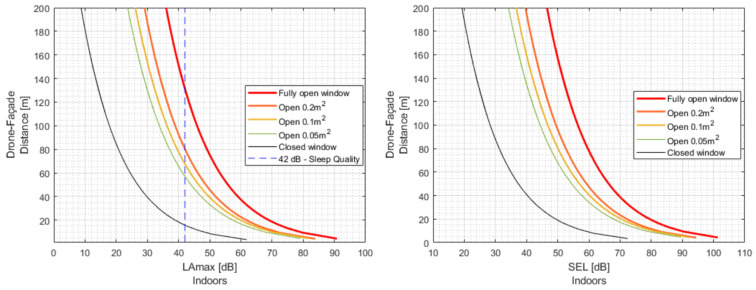
Recommended drone/façade distance for the drone Typhoon based on $LA_{max,indoors}\;$(**left**) and $SEL_{indoors}\;$ (**right**).

**Table 1 ijerph-19-09299-t001:** Overview of Drones Tested by Volpe. Adapted with permission from Ref. [18].

Multirotor Aircraft Model	Number of Rotors	Drone Weight [kg]	MTOW * [kg]	Largest Dimension ** [m]	Aircraft Ground Speed [m/s] (sd ***)	
					Fast	Slow
Gryphon Dynamics **GD28X**	8 (contra-rotating)	11.8	31.7	2.1	13.5 (sd = 0.4)	8.7 (sd = 1.3)
DJI **M200**	4	4.0	6.1	0.9	15.4 (sd = 0.1)	8.6 (sd = 1.6)
Yuneec **Typhoon**	6	1.9	2.4	0.5	12.8 (sd = 0.1)	6.1 (sd = 1.4)

* Maximum take-off weight. ** Rotor tip to rotor tip distance. *** Standard deviation.

**Table 2 ijerph-19-09299-t002:** Estimation of the optimal drone/façade distance DFd for drone GD28X at fast and slow flyover operation near a façade with a conventional window–open area 0.05 m^2^.

Drone	Operation	Glazing Configuration	DFd [m]	Curve Fitting		
**GD28X**	**Flyover**	4-16-4 mm 70 mm PVC internal tilt &turn [32]	Acoustic target: 42 dB $LA_{max,indoors}$ Sleep quality. Waking up in the night and/or too early in the morning [17].	Drone/Façade distance $DFd=a\;e^{-bLA_{max}}$		
				a	b	$R_{adj}^{2}$
	Fast~13.5 m/s	Open area 0.05 m^2^	179.1	2651	−0.0641	0.96
	Slow~8.7 m/s		110.6	1511	−0.0622	0.97

**Table 3 ijerph-19-09299-t003:** Estimation of the optimal drone/façade distance DFd for the tested drones at fast flyover operation near a façade with a conventional window configuration: 4-16-4 mm 70 mm PVC internal tilt &turn–closed.

Drone	Operation	Glazing Configuration	DFd [m]	Curve Fitting		
	Fast Flyover speed	4-16-4 mm 70 mm PVC internal tilt &turn [32]	Acoustic target: 42 dB $LA_{max,indoors}$ Sleep quality. Waking up in the night and/or too early in the morning [17].	Drone/Façade distance $DFd\;=\;a\;e^{-bLA_{max}}$		
				a	b	$R_{adj}^{2}$
GD28X	~13.5 m/s	Closed	62.7	1214	−0.0705	0.95
M200	~15.4 m/s		21.4	439	−0.0718	0.95
Typhoon	~12.8 m/s		15.8	389	−0.0762	0.99

**Table 4 ijerph-19-09299-t004:** Estimation of the optimal drone/façade distance DFd for a drone at fast flyover operation near a façade with a conventional window configuration.

Drone	Operation	Glazing Configuration	DFd [m]	Curve Fitting		
Typhoon	Fast flyover speed [18]	4-16-4 mm 70 mm PVC internal tilt &turn [32]	Acoustic target: 42 dB $LA_{max,indoors}$ Sleep quality. Waking up in the night and/or too early in the morning [17].	Drone/Façade distance $DFd=a\;e^{-bLA_{max}}$		
				a	b	$R_{adj}^{2}$
	~12.8 m/s	Fully open	131.6	2473	−0.0698	0.98
		Open 0.20 m^2^	80.5	1556	−0.0705	0.98
		Open 0.10 m^2^	67.8	1191	−0.0682	0.99
		Open 0.05 m^2^	57.3	1007	−0.0682	0.99
		Closed	15.8	389	−0.0762	0.99

## Data Availability

Part of the audio recordings were obtained from John A. Volpe National Transportation Systems Center. Access to this data can be requested to David R. Read, Christopher Cutler and Juliet Page (John A. Volpe National Transportation Systems Center).

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
