# Peer review of "Requirements for Drone Operations to Minimise Community Noise Impact"

_ijerph, 2022, doi:10.3390/ijerph19159299_

Round 1

Reviewer 1 Report

This paper investigates the problem of setting recommendations for the design of drones operations in an urban context, considering the impact of drones' noise and possible discomfort to the urban community. To this aim, the authors propose a proper modelling framework to set recommendations.

The topic of the article is innovative and worth interest as well as of practical relevance.

The authors could enrich the discussion of the motivation of the work by providing examples of the types of applications and operations that drones are expected to implement in the urban environment. See as example references: Parcel delivery with drones: multi-criteria analysis of trendy system architectures; Consensus-based algorithms for controlling swarms of unmanned aerial vehicles.

The analysis of the acoustic noise of the considered UAV in the urban condition is sound, well structured, and well presented. 

The proposed method is consistently applied to appropriate case studies and the results are discussed in detail.

Author Response

Dear Reviewer,
Thank you for the opportunity to submit a revised version of the manuscript to IJERPH. The authors appreciate the time and effort that you have dedicated to providing your valuable feedback.
We have been able to incorporate changes to reflect all the comments provided. Please see the attached file with the document changes.
Thank you very much for your contribution to the improvement of this paper.

Reviewer 2 Report

General: In my view, this is an important paper which should be published after some amendments. Unfortunately, there seems to have been no final editing of the manuscript. The result is incomplete sentences, unfamiliar grammar to say the least, and indecision in the use of the terms "sound" and "noise". Those readers for whom "noise" is a term of effect ("unwanted sound") will not enjoy the present state of the manuscript.

Details:
L12: "Drone operations have grown considerably over the last few years " This is a very broad statement, and may not be true for all residential areas in the world. (In fact, I observed a decline of drone operations in my residential neighborhood, due to legislative restrictions in the last years).
L 64: criteria (?)
L 73: "Torija, et. al." -> "Torija et al."
L79f: "The inability of existing aircraft noise metrics for accurately capturing drone noise 79 perception, ..." I wonder whether there is any noise metric at all for accurately capturing noise perception. At the end, we are usually confined to rough correspondences between acoustics and human reactions.
L 91: "However, it has been developed so new noise metrics (e.g., based on sound quality parameters) or new evidence for drone noise exposure can be used instead."  Are you sure that this sentence is grammatically correct? Anyway, I don't understand its meaning.
L 114: "thanthe" -> "than the"
L 150f: "This which can be studied.." -> "This can be studied.."?
L 162: "noise level" -> "sound level" (according to 3.1)
L 163: I still insist on a conceptual difference between "sound" and "noise". As far as I understand section 3.1., you describe sound characteristics here, not effects. Therefore all "noise" should be converted to "sound" in section 3.1.
L 210: "on equalduring" -> "on equal during"
L 214: What is mesant by "the drone acoustics noise impact", and what is the relation to annoyance"?. The paper by Beaulieu [30]has a chapter on "Psychoacoustics / Noise Annoyance metrics"; however, it does not measure annoyance in terms of a response, it simply assumes constant relations between annoyance and certain psychoacoustic variables.
L 231: .." frequency of the noise".. What is meant here by "frequency" ? spectral composition of the sound?
L 235: "paths is can be" ??
L 237: "..slow speeds as the. From" ??
L 240: "performanceas" -> "performance as"
L 285: "... based on  government guidance.." which government? UK?
L 364: "The contra-rotating octocopter GD28X was analyzed.."  Which property of the octocopter was analyzed?
L 428 .."while promoting the growth of this new uncrewed transportation system." A paper in a Public Health journal should not "promote" the growth of a certain industry.

Author Response

(The authors gave the same response as above.)

Reviewer 3 Report

With the rapid growth of drone operations, it became an important source of environmental noise. This article resents a modelling framework for setting recommendations for drone operations to minimize community noise impact and illustrates the applicability of the modelling framework through case studies. It provided a preliminary tool for inferring and assessing the impact of drone noise. Specific opinions are as follows.

Line 104: Check the word “vehicles” in the sentence, “The list includes different vehicles’ weights and dimensions.”

In Table 1, add a note to ‘sd’. Dose ‘sd’ represent standard deviation?

In the sentence, “equal to or longer thanthe Slant ”, a blank is missed.

Check this sentence, “This which can be studied as an optimisation parameter on drone infrastructure path planning.”

In 3.1, “possible to obtain the actual sound levels during the drone flyover at the microphonedistance ?. .”, a dot is redundant.

“ie.,” should be replaced by “i.e.,”

on equalduring a fast flyover operation.” should be replaced by “equal during a fast flyover operation.”

The Fig. 5, ??max and ??eq are almost equal during a fast flyover operation. Does this rule apply to all drones or only to the drone “Typhoon”?

Lines 249-252: “R’ and R decreases or dips in the sound reduction index at low (~250Hz) and high-frequency bands (~4000).” However, I can't see this rule at 4000Hz. 

To get the calculated results in Figures 6 and 7, which commercial software was used in this study.  How to take the value of parameters such as the size of the building room or windows and the sound insulation of the building wall?

In the sentence, “frequencies of reduced Sound Reduction performanceas a result of resonance”, a blank is missed.

Check the sentence, “the outdoor ‘fly-over’ procedures at fast and slow speeds as the.”

In Figure 6, does “Shifted reference curve” mean “a fitting curve”?

Lines 271-274:Rewrite this sentence, “the operation of a closer-than-normal noise source may have on an exposed community” 

Lines 331-335: Why did this study only give the results for inward lateral rotation window? It is recommended to give the results for a variety of typical windows.

Author Response

(The authors gave the same response as above.)
